# Early stopping for kernel boosting algorithms: A general analysis with localized complexities

**Yuting Wei**[1]     **Fanny Yang**[2*]     **Martin J. Wainwright**[1,2]
Department of Statistics[1]
Department of Electrical Engineering and Computer Sciences[2]
UC Berkeley
Berkeley, CA 94720
{ytwei, fanny-yang, wainwrig}@berkeley.edu

## Abstract

Early stopping of iterative algorithms is a widely-used form of regularization in statistics, commonly used in conjunction with boosting and related gradient-type algorithms. Although consistency results have been established in some settings, such estimators are less well-understood than their analogues based on penalized regularization. In this paper, for a relatively broad class of loss functions and boosting algorithms (including $L^2$-boost, LogitBoost and AdaBoost, among others), we exhibit a direct connection between the performance of a stopped iterate and the localized Gaussian complexity of the associated function class. This connection allows us to show that local fixed point analysis of Gaussian or Rademacher complexities, now standard in the analysis of penalized estimators, can be used to derive optimal stopping rules. We derive such stopping rules in detail for various kernel classes, and illustrate the correspondence of our theory with practice for Sobolev kernel classes.

## 1   Introduction

While non-parametric models offer great flexibility, they can also lead to overfitting, and thus poor generalization performance. For this reason, procedures for fitting non-parametric models must involve some form of regularization, most commonly done by adding some type of penalty to the objective function. An alternative form of regularization is based on the principle of *early stopping*, in which an iterative algorithm is terminated after a pre-specified number of steps prior to convergence.

While the idea of early stopping is fairly old (e.g., [31, 1, 35]), recent years have witnessed renewed interests in its properties, especially in the context of boosting algorithms and neural network training (e.g., [25, 12]). Over the past decade, a line of work has yielded some theoretical insight into early stopping, including works on classification error for boosting algorithms [3, 13, 18, 23, 39, 40], $L^2$-boosting algorithms for regression [8, 7], and similar gradient algorithms in reproducing kernel Hilbert spaces (e.g. [11, 10, 34, 39, 26]). A number of these papers establish consistency results for particular forms of early stopping, guaranteeing that the procedure outputs a function with statistical error that converges to zero as the sample size increases. On the other hand, there are relatively few results that actually establish *rate optimality* of an early stopping procedure, meaning that the achieved error matches known statistical minimax lower bounds. To the best of our knowledge, Bühlmann and Yu [8] were the first to prove optimality for early stopping of $L^2$-boosting as applied

to spline classes, albeit with a rule that was not computable from the data. Subsequent work by Raskutti et al. [26] refined this analysis of $L^2$-boosting for kernel classes and first established an important connection to the localized Rademacher complexity; see also the related work [39, 27, 9] with rates for particular kernel classes.

More broadly, relative to our rich and detailed understanding of regularization via penalization (e.g., see the books [17, 33, 32, 37] and papers [2, 20] for details), the theory for early stopping regularization is still not as well developed. In particular, for penalized estimators, it is now well-understood that complexity measures such as the *localized Gaussian width*, or its Rademacher analogue, can be used to characterize their achievable rates [2, 20, 32, 37]. Is such a general and sharp characterization also possible in the context of early stopping? The main contribution of this paper is to answer this question in the affirmative for boosting algorithms in regression and classification problems involving functions in reproducing kernel Hilbert spaces (RKHS).

The remainder of this paper is organized as follows. In Section 2, we provide background on boosting methods and reproducing kernel Hilbert spaces, and then introduce the updates studied in this paper. Section 3 is devoted to statements of our main results, followed by a discussion of their consequences for particular function classes in Section 4. We provide simulations that confirm the practical effectiveness of our stopping rules and show close agreement with our theoretical predictions. The proofs for all of our results can be found in the supplemental material.

## 2   Background and problem formulation

The goal of prediction is to learn a function that maps *covariates* $x \in \mathcal{X}$ to *responses* $y \in \mathcal{Y}$. In a regression problem, the responses are typically real-valued, whereas in a classification problem, the responses take values in a finite set. In this paper, we study both regression ($\mathcal{Y} = \mathbb{R}$) and classification problems (e.g., $\mathcal{Y} = \{-1, +1\}$ in the binary case) where we observe a collection of $n$ pairs of the form $\{(x_i, Y_i)\}_{i=1}^n$, with fixed covariates $x_i \in \mathcal{X}$ and corresponding random responses $Y_i \in \mathcal{Y}$ drawn independently from a distribution $\mathbb{P}_{Y|x_i}$. In this section, we provide some necessary background on a gradient-type algorithm which is often referred to as *boosting* algorithm.

### 2.1   Boosting and early stopping

Consider a cost function $\phi : \mathbb{R} \times \mathbb{R} \to [0, \infty)$, where the non-negative scalar $\phi(y, \theta)$ denotes the cost associated with predicting $\theta$ when the true response is $y$. Some common examples of loss functions $\phi$ that we consider in later sections include:

- the *least-squares loss* $\phi(y, \theta) := \frac{1}{2}(y - \theta)^2$ that underlies $L^2$-boosting [8],
- the *logistic regression loss* $\phi(y, \theta) = \ln(1 + e^{-y\theta})$ that underlies the LogitBoost algorithm [14, 15], and
- the *exponential loss* $\phi(y, \theta) = \exp(-y\theta)$ that underlies the AdaBoost algorithm [13].

The least-squares loss is typically used for regression problems (e.g., [8, 11, 10, 34, 39, 26]), whereas the latter two losses are frequently used in the setting of binary classification (e.g., [13, 23, 15]).

Given some loss function $\phi$ and function space $\mathscr{F}$, we define the *population cost functional* $f \mapsto \mathcal{L}(f)$ and the corresponding optimal (minimizing) function[†] via

$$\mathcal{L}(f) := \mathbb{E}_{Y_1^n}\left[\frac{1}{n} \sum_{i=1}^n \phi\big(Y_i, f(x_i)\big)\right], \qquad f^* := \arg\min_{f \in \mathscr{F}} \mathcal{L}(f). \tag{1}$$

Note that with the covariates $\{x_i\}_{i=1}^n$ fixed, the functional $\mathcal{L}$ is a non-random object. As a standard example, when we adopt the least-squares loss $\phi(y, \theta) = \frac{1}{2}(y - \theta)^2$, the population minimizer $f^*$ corresponds to the conditional expectation $x \mapsto \mathbb{E}[Y|x]$. Since we do not have access to the population distribution of the responses however, the computation of $f^*$ is impossible. Given our samples $\{Y_i\}_{i=1}^n$, we consider instead some procedure applied to the *empirical loss*

$$\mathcal{L}_n(f) := \frac{1}{n} \sum_{i=1}^n \phi(Y_i, f(x_i)), \tag{2}$$

---

[†]As clarified in the sequel, our assumptions guarantee uniqueness of $f^*$.

where the population expectation has been replaced by an empirical expectation. For example, when $\mathcal{L}_n$ corresponds to the log likelihood of the samples with $\phi(Y_i, f(x_i)) = \log[\mathbb{P}(Y_i; f(x_i))]$, direct unconstrained minimization of $\mathcal{L}_n$ would yield the maximum likelihood estimator.

It is well-known that direct minimization of $\mathcal{L}_n$ over a rich function class $\mathscr{F}$ may lead to overfitting. A classical method to mitigate this phenomenon is to minimize the sum of the empirical loss with a penalty term. Adjusting the weight on the regularization term allows for trade-off between fit to the data, and some form of regularity or smoothness of the fit. The behavior of such penalized estimation methods is quite well understood (see e.g. the books [17, 33, 32, 37] and papers [2, 20] for details).

In this paper, we study a form of *algorithmic regularization*, based on applying a gradient-type algorithm to $\mathcal{L}_n$. In particular, we consider *boosting algorithms* (see survey paper [7]) which involve "boosting" or improve the fit of a function via a sequence of additive updates (see e.g. [28, 13, 6, 5, 29]) and can be understood as forms of functional gradient methods [23, 15]. Instead of running until convergence, we then stop it "early"—that is, after some fixed number of steps. The way in which the number of steps is chosen is referred to as a stopping rule, and the overall procedure is referred to as *early stopping* of a boosting algorithm.

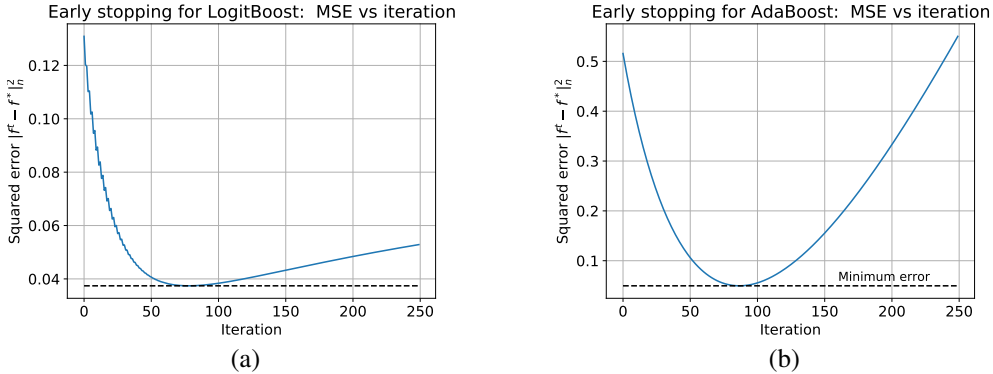

Figure 1: Plots of the squared error $\|f^t - f^*\|_n^2 = \frac{1}{n}\sum_{i=1}^n (f^t(x_i) - f^*(x_i))^2$ versus the iteration number $t$ for (a) LogitBoost using a first-order Sobolev kernel (b) AdaBoost using the same first-order Sobolev kernel $\mathbb{K}(x, x') = 1 + \min(x, x')$ which generates a class of Lipschitz functions (splines of order one). Both plots correspond to a sample size $n = 100$.

In more detail, a broad class of boosting algorithms [23] generate a sequence $\{f^t\}_{t=0}^{\infty}$ via updates of the form

$$f^{t+1} = f^t - \alpha^t g^t \quad \text{with} \quad g^t \propto \underset{\|d\|_{\mathscr{F}} \leq 1}{\arg\max} \langle \nabla \mathcal{L}_n(f^t), d(x_1^n) \rangle, \tag{3}$$

where the scalar $\{\alpha^t\}_{t=0}^{\infty}$ is a sequence of step sizes chosen by the user, the constraint $\|d\|_{\mathscr{F}} \leq 1$ defines the unit ball in a given function class $\mathscr{F}$, $\nabla \mathcal{L}_n(f) \in \mathbb{R}^n$ denotes the gradient taken at the vector $(f(x_1), \ldots, f(x_n))$, and $\langle h, g \rangle$ is the usual inner product between vectors $h, g \in \mathbb{R}^n$. For non-decaying step sizes and a convex objective $\mathcal{L}_n$, running this procedure for an infinite number of iterations will lead to a minimizer of the empirical loss, thus causing overfitting. In order to illustrate this phenomenon, Figure 1 provides plots of the squared error $\|f^t - f^*\|_n^2 := \frac{1}{n}\sum_{i=1}^n (f^t(x_i) - f^*(x_i))^2$ versus the iteration number, for LogitBoost in panel (a) and AdaBoost in panel (b). (See Section 4.2 for more details on how these experiments were set up.)

In these plots, the dotted line indicates the minimum mean-squared error $\rho_n^2$ over all iterates of that particular run of the algorithm. Both plots are qualitatively similar, illustrating the existence of a "good" number of iterations to take, after which the MSE greatly increases. Hence a natural problem is to decide at what iteration $T$ to stop such that the iterate $f^T$ satisfies bounds of the form

$$\mathcal{L}(f^T) - \mathcal{L}(f^*) \precsim \rho_n^2 \quad \text{and} \quad \|f^T - f^*\|_n^2 \precsim \rho_n^2 \tag{4}$$

with high probability. The main results of this paper provide a stopping rule $T$ for which bounds of the form (4) do in fact hold with high probability over the randomness in the observed responses.

Moreover, as shown by our later results, under suitable regularity conditions, the expectation of the minimum squared error $\rho_n^2$ is proportional to the *statistical minimax risk* $\inf_{\hat{f}} \sup_{f \in \mathcal{F}} \mathbb{E}[\mathcal{L}(\hat{f}) - \mathcal{L}(f)]$, where the infimum is taken over all possible estimators $\hat{f}$. Coupled with our stopping time guarantee (4) this implies that our estimate achieves the minimax risk up to constant factors. As a result, our bounds are unimprovable in general (see Corollary 1).

## 2.2 Reproducing Kernel Hilbert Spaces

The analysis of this paper focuses on algorithms with the update (3) when the function class $\mathcal{F}$ is a reproducing kernel Hilbert space $\mathcal{H}$ (RKHS, see standard sources [36, 16, 30, 4]), consisting of functions mapping a domain $\mathcal{X}$ to the real line $\mathbb{R}$. Any RKHS is defined by a bivariate symmetric *kernel function* $\mathbb{K} : \mathcal{X} \times \mathcal{X} \to \mathbb{R}$ which is required to be positive semidefinite, i.e. for any integer $N \geq 1$ and a collection of points $\{x_j\}_{j=1}^{N}$ in $\mathcal{X}$, the matrix $[\mathbb{K}(x_i, x_j)]_{ij} \in \mathbb{R}^{N \times N}$ is positive semidefinite. The associated RKHS is the closure of linear span of the form $f(\cdot) = \sum_{j \geq 1} \omega_j \mathbb{K}(\cdot, x_j)$, where $\{x_j\}_{j=1}^{\infty}$ is some collection of points in $\mathcal{X}$, and $\{\omega_j\}_{j=1}^{\infty}$ is a real-valued sequence. For two functions $f_1, f_2 \in \mathcal{H}$ which can be expressed as a finite sum $f_1(\cdot) = \sum_{i=1}^{\ell_1} \alpha_i \mathbb{K}(\cdot, x_i)$ and $f_2(\cdot) = \sum_{j=1}^{\ell_2} \beta_j \mathbb{K}(\cdot, x_j)$, the inner product is defined as $\langle f_1, f_2 \rangle_{\mathcal{H}} = \sum_{i=1}^{\ell_1} \sum_{j=1}^{\ell_2} \alpha_i \beta_j \mathbb{K}(x_i, x_j)$ with induced norm $\|f_1\|_{\mathcal{H}}^2 = \sum_{i=1}^{\ell_1} \alpha_i^2 \mathbb{K}(x_i, x_i)$. For each $x \in \mathcal{X}$, the function $\mathbb{K}(\cdot, x)$ belongs to $\mathcal{H}$, and satisfies the reproducing relation $\langle f, \mathbb{K}(\cdot, x) \rangle_{\mathcal{H}} = f(x)$ for all $f \in \mathcal{H}$.

Throughout this paper, we assume that the kernel function is uniformly bounded, meaning that there is a constant $L$ such that $\sup_{x \in \mathcal{X}} \mathbb{K}(x, x) \leq L$. Such a boundedness condition holds for many kernels used in practice, including the Gaussian, Laplacian, Sobolev, other types of spline kernels, as well as any trace class kernel with trignometric eigenfunctions. By rescaling the kernel as necessary, we may assume without loss of generality that $L = 1$. As a consequence, for any function $f$ such that $\|f\|_{\mathcal{H}} \leq r$, we have by the reproducing relation that

$$\|f\|_{\infty} = \sup_x \langle f, \mathbb{K}(\cdot, x) \rangle_{\mathcal{H}} \leq \|f\|_{\mathcal{H}} \sup_x \|\mathbb{K}(\cdot, x)\|_{\mathcal{H}} \leq r.$$

Given samples $\{(x_i, y_i)\}_{i=1}^{n}$, by the representer theorem [19], it is sufficient to restrict ourselves to the linear subspace $\mathcal{H}_n = \overline{\text{span}}\{\mathbb{K}(\cdot, x_i)\}_{i=1}^{n}$, for which all $f \in \mathcal{H}_n$ can be expressed as

$$f = \frac{1}{\sqrt{n}} \sum_{i=1}^{n} \omega_i \mathbb{K}(\cdot, x_i) \tag{5}$$

for some coefficient vector $\omega \in \mathbb{R}^n$. Among those functions which achieve the infimum in expression (1), let us define $f^*$ as the one with the minimum Hilbert norm. This definition is equivalent to restricting $f^*$ to be in the linear subspace $\mathcal{H}_n$.

## 2.3 Boosting in kernel spaces

For a finite number of covariates $x_i$ from $i = 1 \ldots n$, let us define the *normalized kernel matrix* $K \in \mathbb{R}^{n \times n}$ with entries $K_{ij} = \mathbb{K}(x_i, x_j)/n$. Since we can restrict the minimization of $\mathcal{L}_n$ and $\mathcal{L}$ from $\mathcal{H}$ to the subspace $\mathcal{H}_n$ w.l.o.g., using expression (5) we can then write the function value vectors $f(x_1^n) := (f(x_1), \ldots, f(x_n))$ as $f(x_1^n) = \sqrt{n} K \omega$. As there is a one-to-one correspondence between the $n$-dimensional vectors $f(x_1^n) \in \mathbb{R}^n$ and the corresponding function $f \in \mathcal{H}_n$ in $\mathcal{H}$ by the representer theorem, minimization of an empirical loss in the subspace $\mathcal{H}_n$ essentially becomes the $n$-dimensional problem of fitting a response vector $y$ over the set $\text{range}(K)$. In the sequel, all updates will thus be performed on the function value vectors $f(x_1^n)$.

With a change of variable $d(x_1^n) = \sqrt{n}\sqrt{K}z$ we then have $d^t(x_1^n) := \arg\max_{\|d\|_{\mathcal{H}} \leq 1} \langle \nabla \mathcal{L}_n(f^t), d(x_1^n) \rangle = \frac{\sqrt{n} K \nabla \mathcal{L}_n(f^t)}{\sqrt{\nabla \mathcal{L}_n(f^t) K \nabla \mathcal{L}_n(f^t)}}$, where the maximum is taken over vectors $d \in \text{range}(K)$. In this paper we study $g^t = \langle \nabla \mathcal{L}_n(f^t), d^t(x_1^n) \rangle d^t$ in the boosting update (3), so that the function value iterates take the form

$$f^{t+1}(x_1^n) = f^t(x_1^n) - \alpha n K \nabla \mathcal{L}_n(f^t), \tag{6}$$

where $\alpha > 0$ is a constant stepsize choice. Choosing $f^0(x_1^n) = 0$ ensures that all iterates $f^t(x_1^n)$ remain in the range space of $K$. Our goal is to propose a stopping time $T$ such that the averaged function $\widehat{f} = \frac{1}{T}\sum_{t=1}^T f^t$ satisfies bounds of the type (4). Importantly, we exhibit such bounds with a statistical error term $\delta_n$ that is specified by the *localized Gaussian complexity* of the kernel class.

## 3 Main results

We now turn to the statement of our main results, beginning with the introduction of some regularity assumptions.

### 3.1 Assumptions

Recall from our earlier set-up that we differentiate between the empirical loss function $\mathcal{L}_n$ in expression (2), and the population loss $\mathcal{L}$ in expression (1). Apart from assuming differentiability of both functions, all of our remaining conditions are imposed on the population loss. Such conditions at the population level are weaker than their analogues at the empirical level.

For a given radius $r > 0$, let us define the Hilbert ball around the optimal function $f^*$ as

$$\mathbb{B}_{\mathscr{H}}(f^*, r) := \{f \in \mathscr{H} \mid \|f - f^*\|_{\mathscr{H}} \le r\}. \tag{7}$$

Our analysis makes particular use of this ball defined for the radius $C_{\mathscr{H}}^2 := 2\max\{\|f^*\|_{\mathscr{H}}^2,\, 32,\, \sigma^2\}$, where $\sigma$ is the effective noise level defined as

$$\sigma := \begin{cases} \min\left\{t \mid \max_{i=1,\ldots,n} \mathbb{E}[e^{((Y_i - f^*(x_i))^2/t^2)}] < \infty\right\} & \text{for least squares} \\ 4\,(2M+1)(1 + 2C_{\mathscr{H}}) & \text{for } \phi'\text{-bounded losses.} \end{cases} \tag{8}$$

We assume that the population loss is $m$-strongly convex and $M$-smooth over $\mathbb{B}_{\mathscr{H}}(f^*, 2C_{\mathscr{H}})$, meaning that the sandwich inequality

$$m\text{-}M\text{-condition} \quad \frac{m}{2}\|f - g\|_n^2 \le \mathcal{L}(f) - \mathcal{L}(g) - \langle \nabla\mathcal{L}(g),\, f(x_1^n) - g(x_1^n)\rangle \le \frac{M}{2}\|f - g\|_n^2$$

holds for all $f, g \in \mathbb{B}_{\mathscr{H}}(f^*, 2C_{\mathscr{H}})$. On top of that we assume $\phi$ to be $M$-Lipschitz in the second argument. To be clear, here $\nabla\mathcal{L}(g)$ denotes the vector in $\mathbb{R}^n$ obtained by taking the gradient of $\mathcal{L}$ with respect to the vector $g(x_1^n)$. It can be verified by a straightforward computation that when $\mathcal{L}$ is induced by the least-squares cost $\phi(y, \theta) = \frac{1}{2}(y - \theta)^2$, the $m$-$M$-condition holds for $m = M = 1$. The logistic and exponential loss satisfy this condition (see supp. material), where it is key that we have imposed the condition *only locally* on the ball $\mathbb{B}_{\mathscr{H}}(f^*, 2C_{\mathscr{H}})$.

In addition to the least-squares cost, our theory also applies to losses $\mathcal{L}$ induced by scalar functions $\phi$ that satisfy the following condition:

$$\phi'\text{-boundedness} \quad \max_{i=1,\ldots,n}\left|\frac{\partial\phi(y,\theta)}{\partial\theta}\right|_{\theta = f(x_i)} \le B \qquad \text{for all } f \in \mathbb{B}_{\mathscr{H}}(f^*, 2C_{\mathscr{H}}) \text{ and } y \in \mathcal{Y}.$$

This condition holds with $B = 1$ for the logistic loss for all $\mathcal{Y}$, and $B = \exp(2.5C_{\mathscr{H}})$ for the exponential loss for binary classification with $\mathcal{Y} = \{-1, 1\}$, using our kernel boundedness condition. Note that whenever this condition holds with some finite $B$, we can always rescale the scalar loss $\phi$ by $1/B$ so that it holds with $B = 1$, and we do so in order to simplify the statement of our results.

### 3.2 Upper bound in terms of localized Gaussian width

Our upper bounds involve a complexity measure known as the localized Gaussian width. In general, Gaussian widths are widely used to obtain risk bounds for least-squares and other types of $M$-estimators. In our case, we consider Gaussian complexities for "localized" sets of the form

$$\mathcal{E}_n(\delta, 1) := \left\{f - g \mid f, g \in \mathscr{H},\quad \|f - g\|_{\mathscr{H}} \le 1,\quad \|f - g\|_n \le \delta\right\}. \tag{9}$$

The Gaussian complexity localized at scale $\delta$ is given by

$$\mathcal{G}_n\big(\mathcal{E}_n(\delta, 1)\big) := \mathbb{E}\left[\sup_{g \in \mathcal{E}_n(\delta,1)} \frac{1}{n}\sum_{i=1}^n w_i g(x_i)\right], \tag{10}$$

where $(w_1, \ldots, w_n)$ denotes an i.i.d. sequence of standard Gaussian variables.

An essential quantity in our theory is specified by a certain fixed point equation that is now standard in empirical process theory [32, 2, 20, 26]. The *critical radius* $\delta_n$ is the smallest positive scalar such that

$$\frac{\mathcal{G}_n(\mathcal{E}_n(\delta, 1))}{\delta} \leq \frac{\delta}{\sigma}. \tag{11}$$

We note that past work on localized Rademacher and Gaussian complexity [24, 2] guarantee that there exists a unique $\delta_n > 0$ that satisfies this condition, so that our definition is sensible.

### 3.2.1 Upper bounds on excess risk and empirical $L^2(\mathbb{P}_n)$-error

With this set-up, we are now equipped to state our main theorem. It provides high-probability bounds on the excess risk and $L^2(\mathbb{P}_n)$-error of the estimator $\bar{f}^T := \frac{1}{T} \sum_{t=1}^T f^t$ defined by averaging the $T$ iterates of the algorithm.

**Theorem 1.** *Consider any loss function satisfying the $m$-$M$-condition and the $\phi'$-boundedness condition (if not least squares), for which we generate function iterates $\{f^t\}_{t=0}^{\infty}$ of the form* (6) *with step size* $\alpha \in (0, \min\{\frac{1}{M}, M\}]$*, initialized at $f^0 = 0$. Then, if $n$ is large enough such that $\delta_n \leq \frac{M}{m}$, for all iterations $T = 0, 1, \ldots \lfloor \frac{m}{8M\delta_n^2} \rfloor$, the averaged function estimate $\bar{f}^T$ satisfies the bounds*

$$\mathcal{L}(\bar{f}^T) - \mathcal{L}(f^*) \leq CM\Big(\frac{1}{\alpha m T} + \frac{\delta_n^2}{m^2}\Big), \quad \text{and} \tag{12a}$$

$$\|\bar{f}^T - f^*\|_n^2 \leq C\Big(\frac{1}{\alpha m T} + \frac{\delta_n^2}{m^2}\Big), \tag{12b}$$

*where both inequalities hold with probability at least $1 - c_1 \exp(-C_2 \frac{m^2 n \delta_n^2}{\sigma^2})$.*

In our statements, constants of the form $c_j$ are universal, whereas capital $C_j$ may depend on parameters of the joint distribution and population loss $\mathcal{L}$. In the previous theorem, $C_2 = \{\frac{m^2}{\sigma^2}, 1\}$ and $C$ depends on the squared radius $C_{\mathscr{H}}^2 := 2\max\{\|f^*\|_{\mathscr{H}}^2, 32, \sigma^2\}$. In order to gain intuition for the claims in the theorem, note that (disregarding factors depending on $(m, M)$), for all iterations $T \lesssim 1/\delta_n^2$, the first term $\frac{1}{\alpha m T}$ dominates the second term $\frac{\delta_n^2}{m^2}$, so that taking further iterations reduces the upper bound on the error until $T \sim 1/\delta_n^2$, at which point the upper bound on the error is of the order $\delta_n^2$.

Furthermore, note that similar bounds as in Theorem 1 can be obtained for the expected loss (over the response $y_i$, with the design fixed) by a simple integration argument. Hence if we perform updates with step size $\alpha = \frac{1}{M}$, after $\tau := \frac{m}{\delta_n^2 \max\{8, M\}}$ iterations, the mean squared error is bounded as

$$\mathbb{E}\|\bar{f}^\tau - f^*\|_n^2 \leq C' \frac{\delta_n^2}{m^2}, \tag{13}$$

where we use $M \geq m$ and where $C'$ is another constant depending on $C_{\mathscr{H}}$. It is worth noting that guarantee (13) matches the best known upper bounds for kernel ridge regression (KRR)—indeed, this must be the case, since a sharp analysis of KRR is based on the same notion of localized Gaussian complexity. Thus, our results establish a strong parallel between the *algorithmic regularization* of early stopping, and the *penalized regularization* of kernel ridge regression. Moreover, as discussed in Section 3.3, under suitable regularity conditions on the RKHS, the critical squared radius $\delta_n^2$ also acts as a lower bound for the expected risk, i.e. our upper bounds are not improvable in general.

Compared with the work of Raskutti et al. [26], which also analyzes the kernel boosting iterates of the form (6), our theory more directly analyzes the effective function class that is explored in the boosting process by taking $T$ steps, with the localized Gaussian width (10) appearing more naturally. In addition, our analysis applies to a broader class of loss functions beyond least-squares.

In the case of reproducing kernel Hilbert spaces, it is possible to sandwich the localized Gaussian complexity by a function of the eigenvalues of the kernel matrix. Mendelson [24] provides this argument in the case of the localized Rademacher complexity, but similar arguments apply to the

localized Gaussian complexity. Letting $\mu_1 \geq \mu_2 \geq \cdots \geq \mu_n \geq 0$ denote the ordered eigenvalues of the normalized kernel matrix $K$, define the function

$$\mathcal{R}(\delta) = \frac{1}{\sqrt{n}}\sqrt{\sum_{j=1}^{n}\min\{\delta^2, \mu_j\}}. \tag{14}$$

Up to a universal constant, this function is an upper bound on the Gaussian width $\mathcal{G}_n\big(\mathcal{E}_n(\delta, 1)\big)$ for all $\delta \geq 0$, and up to another universal constant, it is also a lower bound for all $\delta \geq \frac{1}{\sqrt{n}}$.

Note that the critical radius $\delta_n^2$ only depends on our observations $\{(x_i, y_i)\}_{i=1}^n$ through the solution of inequality (11). In many cases, with examples given in Section 4, it is possible to compute or upper bound this critical radius, so that a concrete stopping rule can indeed by calculated in advance.

### 3.3 Achieving minimax lower bounds

We claim that when the noise $Y - f(x)$ is Gaussian, for a broad class of kernels, upper bound (13) matches the known minimax lower bound, thus is unimprovable in general. In particular, Yang et al. [38] define the class of *regular kernels*, which includes the Gaussian and Sobolev kernels as particular cases. For such kernels, the authors provide a minimax lower bound over the unit ball of the Hilbert space involving $\delta_n$, which implies that any estimator $\widehat{f}$ has prediction risk lower bounded as

$$\sup_{\|f^*\|_{\mathscr{H}} \leq 1} \mathbb{E}\|\widehat{f} - f^*\|_n^2 \geq c_\ell \delta_n^2. \tag{15}$$

Comparing the lower bound (15) with upper bound (13) for our estimator $\bar{f}^T$ stopped after $O(1/\delta_n^2)$ many steps, it follows that the bounds proven in Theorem 1 are unimprovable apart from constant factors. We summarize our findings in the following corollary:

**Corollary 1.** *For the class of regular kernels and any function $f^*$ with $\|f^*\|_{\mathscr{H}} \leq 1$, running $T := \lfloor \frac{1}{\delta_n^2 \max\{8, M\}} \rfloor$ iterations with step size $\alpha = \frac{m}{M}$ and $f^0 = 0$ yields an estimate $\bar{f}^T$ such that*

$$\mathbb{E}\|\bar{f}^T - f^*\|_n^2 \asymp \inf_{\widehat{f}} \sup_{\|f^*\|_{\mathscr{H}} \leq 1} \mathbb{E}\|\widehat{f} - f^*\|_n^2, \tag{16}$$

*where the infimum is taken over all measurable functions of the input data and the expectation is taken over the randomness of the response variables $\{Y_i\}_{i=1}^n$.*

On a high level, the statement in Corollary 1 implies that stopping early essentially prevents us from overfitting to the data and automatically finds the optimal balance between low training error (i.e. fitting the data well) and low model complexity (i.e. generalizing well).

## 4 Consequences for various kernel classes

In this section, we apply Theorem 1 to derive some concrete rates for different kernel spaces and then illustrate them with some numerical experiments. It is known that the complexity of a RKHS in association with fixed covariates $\{x_i\}_{i=1}^n$ can be characterized by the decay rate of the eigenvalues $\{\mu_j\}_{j=1}^n$ of the normalized kernel matrix $K$. The representation power of a kernel class is directly correlated with the eigen-decay: the faster the decay, the smaller the function class.

### 4.1 Theoretical predictions as a function of decay

In this section, let us consider two broad types of eigen-decay:

- $\gamma$-**exponential decay**: For some $\gamma > 0$, the kernel matrix eigenvalues satisfy a decay condition of the form $\mu_j \leq c_1 \exp(-c_2 j^\gamma)$, where $c_1, c_2$ are universal constants. Examples of kernels in this class include the Gaussian kernel, which for the Lebesgue measure satisfies such a bound with $\gamma = 2$ (real line) or $\gamma = 1$ (compact domain).

- $\beta$-**polynomial decay**: For some $\beta > 1/2$, the kernel matrix eigenvalues satisfy a decay condition of the form $\mu_j \leq c_1 j^{-2\beta}$, where $c_1$ is a universal constant. Examples of kernels in this class

include the $k^{th}$-order Sobolev spaces for some fixed integer $k \geq 1$ with Lebesgue measure on a bounded domain. We consider Sobolev spaces that consist of functions that have $k^{th}$-order weak derivatives $f^{(k)}$ being Lebesgue integrable and $f(0) = f^{(1)}(0) = \cdots = f^{(k-1)}(0) = 0$. For such classes, the $\beta$-polynomial decay condition holds with $\beta = k$.

Given eigendecay conditions of these types, it is possible to compute an upper bound on the critical radius $\delta_n$. In particular, using the fact that the function $\mathcal{R}$ from equation (14) is an upper bound on the function $\mathcal{G}_n(\mathcal{E}(\delta, 1))$, we can show that for $\gamma$-exponentially decaying kernels, we have $\delta_n^2 \precsim \frac{(\log n)^{1/\gamma}}{n}$, whereas for $\beta$-polynomial kernels, we have $\delta_n^2 \precsim n^{-\frac{2\beta}{2\beta+1}}$ up to universal constants. Combining with our Theorem 1, we obtain the following result:

**Corollary 2** (Bounds based on eigendecay). *Suppose we apply boosting with stepsize $\alpha = \frac{m}{M}$ and initialization $f^0 = 0$ on the empirical loss function $\mathcal{L}_n$ which satisfies the $m$-$M$-condition and $\phi'$-boundedness conditions, and is defined on covariate-response pairs $\{(x_i, Y_i)\}_{i=1}^n$ with $Y_i$ drawn from the distribution $\mathbb{P}_{Y|x_i}$. Then, the error of the averaged iterate $\bar{f}^T$ satisfies the following upper bounds with high probability, "$\precsim$" neglecting dependence on problem parameters such as $(m, M)$:*

*(a) For kernels with $\gamma$-exponential eigen-decay with respect to $\{x_i\}_{i=1}^n$:*
$$\|\bar{f}^T - f^*\|_n^2 \lesssim \frac{\log^{1/\gamma} n}{n} \text{ when stopped after } T \asymp \frac{n}{\log^{1/\gamma} n} \text{ steps.}$$

*(b) For kernels with $\beta$-polynomial eigen-decay with respect to $\{x_i\}_{i=1}^n$:*
$$\|\bar{f}^T - f^*\|_n^2 \lesssim n^{-2\beta/(2\beta+1)}, \text{ when stopped after } T \asymp n^{2\beta/(2\beta+1)} \text{ steps.}$$

*In particular, these bounds hold for LogitBoost and AdaBoost.*

To the best of our knowledge, this result is the first to show non-asymptotic and optimal statistical rates for the $\|\cdot\|_n^2$-error when using early stopping LogitBoost or AdaBoost with an explicit dependence of the stopping rule on $n$. Our results also yield similar guarantees for $L^2$-boosting, as has been established in past work [26]. Note that we can observe a similar trade-off between computational efficiency and statistical accuracy as in the case of kernel least-squares regression [39, 26]: although larger kernel classes (e.g. Sobolev classes) yield higher estimation errors, boosting updates reach the optimum faster than for a smaller kernel class (e.g. Gaussian kernels).

### 4.2 Numerical experiments

We now describe some numerical experiments that provide illustrative confirmations of our theoretical predictions using the first-order Sobolev kernel as a typical example for kernel classes with polynomial eigen-decay. In particular, we consider the first-order Sobolev space of Lipschitz functions on the unit interval $[0, 1]$, defined by the kernel $\mathbb{K}(x, x') = 1 + \min(x, x')$, and with the design points $\{x_i\}_{i=1}^n$ set equidistantly over $[0, 1]$. Note that the equidistant design yields $\beta$-polynomial decay of the eigenvalues of $K$ with $\beta = 1$ so that $\delta_n^2 \asymp n^{-2/3}$. Accordingly, our theory predicts that the stopping time $T = (cn)^{2/3}$ should lead to an estimate $\bar{f}^T$ such that $\|\bar{f}^T - f^*\|_n^2 \precsim n^{-2/3}$.

In our experiments for $L^2$-Boost, we sampled $Y_i$ according to $Y_i = f^*(x_i) + w_i$ with $w_i \sim \mathcal{N}(0, 0.5)$, which corresponds to the probability distribution $\mathbb{P}(Y \mid x_i) = \mathcal{N}(f^*(x_i); 0.5)$, where $f^*(x) = |x - \frac{1}{2}| - \frac{1}{4}$ is defined on the unit interval $[0, 1]$. By construction, the function $f^*$ belongs to the first-order Sobolev space with $\|f^*\|_{\mathscr{H}} = 1$. For LogitBoost, we sampled $Y_i$ according to $\text{Bern}(p(x_i))$ where $p(x) = \frac{\exp(f^*(x))}{1+\exp(f^*(x))}$ with the same $f^*$. We chose $f^0 = 0$ in all cases, and ran the updates (6) for $L^2$-Boost and LogitBoost with the constant step size $\alpha = 0.75$. We compared various stopping rules to the *oracle gold standard $G$*, which chooses the stopping time $G = \arg\min_{t \geq 1} \|f^t - f^*\|_n^2$ that yields the minimum prediction error among all iterates $\{f^t\}$. Although this procedure is unimplementable in practice, but it serves as a convenient lower bound with which to compare. Figure 2 shows plots of the mean-squared error $\|\bar{f}^T - f^*\|_n^2$ over the sample size $n$ averaged over 40 trials, for the gold standard $T = G$ and stopping rules based on $T = (7n)^\kappa$ for different choices of $\kappa$. Error bars correspond to the standard errors computed from our simulations. Panel (a) shows the behavior for $L^2$-boosting, whereas panel (b) shows the behavior for LogitBoost.

Note that both plots are qualitatively similar and that the theoretically derived stopping rule $T = (7n)^\kappa$ with $\kappa^* = 2/3 = 0.67$, while slightly worse than the Gold standard, tracks its performance closely.

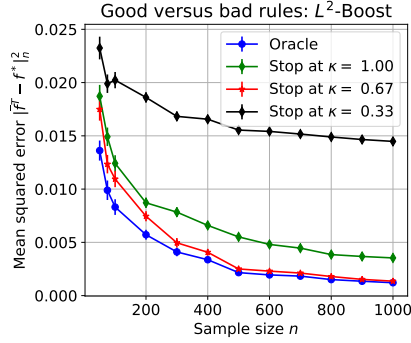
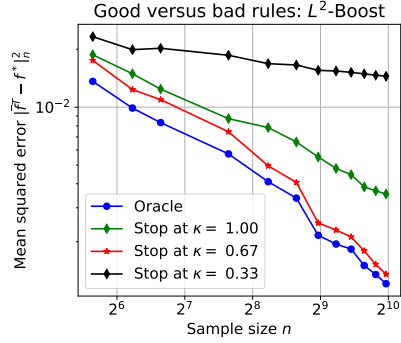

|  (a) | (b) |

Figure 2: The mean-squared errors for the stopped iterates $\bar{f}^T$ at the Gold standard, i.e. iterate with the minimum error among all unstopped updates (blue) and at $T = (7n)^{\kappa}$ (with the theoretically optimal $\kappa = 0.67$ in red, $\kappa = 0.33$ in black and $\kappa = 1$ in green) for (a) $L^2$-Boost and (b) LogitBoost.

We also performed simulations for some "bad" stopping rules, in particular for an exponent $\kappa$ *not equal* to $\kappa^* = 2/3$, indicated by the green and black curves. In the log scale plots in Figure 3 we can clearly see that for $\kappa \in \{0.33, 1\}$ the performance is indeed much worse, with the difference in slope even suggesting a different scaling of the error with the number of observations $n$. Recalling our discussion for Figure 1, this phenomenon likely occurs due to underfitting and overfitting effects.

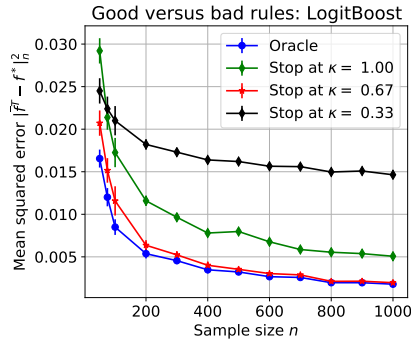
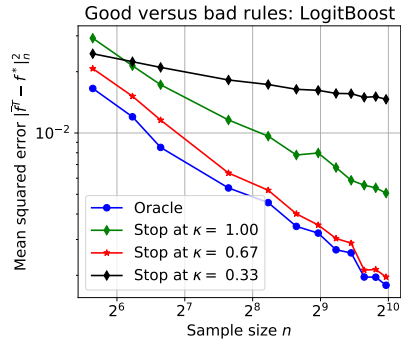

|  (a) | (b) |

Figure 3: Logarithmic plots of the mean-squared errors at the Gold standard in blue and at $T = (7n)^{\kappa}$ (with the theoretically optimal rule for $\kappa = 0.67$ in red, $\kappa = 0.33$ in black and $\kappa = 1$ in green) for (a) $L^2$-Boost and (b) LogitBoost.

## 5 Discussion

In this paper, we have proven non-asymptotic bounds for early stopping of kernel boosting for a relatively broad class of loss functions. These bounds allowed us to propose simple stopping rules which, for the class of regular kernel functions [38], yield minimax optimal rates of estimation. Although the connection between early stopping and regularization has long been studied and explored in the literature, to the best of our knowledge, this paper is the first one to establish a general relationship between the statistical optimality of stopped iterates and the localized Gaussian complexity, a quantity well-understood to play a central role in controlling the behavior of regularized estimators based on penalization [32, 2, 20, 37].

There are various open questions suggested by our results. Can fast approximation techniques for kernels be used to approximately compute optimal stopping rules without having to calculate all eigenvalues of the kernel matrix? Furthermore, we suspect that similar guarantees can be shown for the stopped estimator $f^T$ which we observed to behave similarly to the averaged estimator $\bar{f}^T$ in our simulations. It would be of interest to establish results on $f^T$ directly.

## Acknowledgements

This work was partially supported by DOD Advanced Research Projects Agency W911NF-16-1-0552, National Science Foundation grant NSF-DMS-1612948, and Office of Naval Research Grant DOD-ONR-N00014.

## Footnotes

*Yuting Wei and Fanny Yang contributed equally to this work.

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
