[Supplementary Material · Nips_Camera_Supp.pdf]

# A  Proof of main results

In this section, we present the proofs of our main results. The technical details are deferred to Section B.

In the following, recalling the discussion in Section 2.3, we denote the vector of function values of a function $f \in \mathscr{H}$ evaluated at $(x_1, x_2, \ldots, x_n)$ as $\theta_f := f(x_1^n) = (f(x_1), f(x_2), \ldots f(x_n)) \in \mathbb{R}^n$, where we omit the subscript $f$ when it is clear from the context. As mentioned in the main text, updates on the function value vectors $\theta^t \in \mathbb{R}^n$ correspond uniquely to updates of the functions $f^t \in \mathscr{H}$. In the following we repeatedly abuse notation by defining the Hilbert norm and empirical norm on vectors in $\Delta \in \mathrm{range}(K)$ as

$$\|\Delta\|_{\mathscr{H}}^2 = \frac{1}{n} \Delta^T K^\dagger \Delta \quad \text{and} \quad \|\Delta\|_n^2 = \frac{1}{n}\|\Delta\|_2^2,$$

where $K^\dagger$ is the pseudoinverse of $K$. We also use $\mathbb{B}_{\mathscr{H}}(\theta, r)$ to denote the ball with respect to the $\|\cdot\|_{\mathscr{H}}$-norm in $\mathrm{range}(K)$.

## A.1  Proof of Theorem 1

The proof of our main theorem is based on a sequence of lemmas, all of which are stated with the assumptions of Theorem 1 in force. The first lemma establishes a bound on the empirical norm $\|\cdot\|_n$ of the error $\Delta^{t+1} := \theta^{t+1} - \theta^*$, provided that its Hilbert norm is suitably controlled.

**Lemma 1.** *For any stepsize $\alpha \in (0, \frac{1}{M}]$ and any iteration $t$ we have*

$$\frac{m}{2}\|\Delta^{t+1}\|_n^2 \le \frac{1}{2\alpha}\left\{\|\Delta^t\|_{\mathscr{H}}^2 - \|\Delta^{t+1}\|_{\mathscr{H}}^2\right\} + \langle \nabla\mathcal{L}(\theta^* + \Delta^t) - \nabla\mathcal{L}_n(\theta^* + \Delta^t), \Delta^{t+1}\rangle. \quad (17)$$

See Section B.1 for the proof of this claim.

The second term on the right-hand side of the bound (17) involves the difference between the population and empirical gradient operators. Since this difference is being evaluated at the random points $\Delta^t$ and $\Delta^{t+1}$, the following lemma establishes a form of uniform control on this term.

Let us define the set

$$\mathbb{S} := \left\{ \Delta, \widetilde{\Delta} \in \mathbb{R}^n \mid \|\Delta\|_{\mathscr{H}} \ge 1, \quad \text{and} \quad \theta^* + \Delta, \ \theta^* + \widetilde{\Delta} \in \mathbb{B}_{\mathscr{H}}(\theta^*, 2C_{\mathscr{H}}) \right\}, \quad (18)$$

and consider the uniform bound

$$\langle \nabla\mathcal{L}(\theta^* + \widetilde{\Delta}) - \nabla\mathcal{L}_n(\theta^* + \widetilde{\Delta}), \Delta\rangle \le 2\delta_n\|\Delta\|_n + 2\delta_n^2\|\Delta\|_{\mathscr{H}} + \frac{m}{c_3}\|\Delta\|_n^2 \quad \text{for all } \Delta, \widetilde{\Delta} \in \mathbb{S}. \quad (19)$$

**Lemma 2.** *Let $\mathcal{E}$ be the event that bound (19) holds. There are universal constants $(c_1, c_2)$ such that $\mathbb{P}[\mathcal{E}] \ge 1 - c_1 \exp(-c_2 \frac{m^2 n \delta_n^2}{\sigma^2})$.*

See Section B.2 for the proof of Lemma 2.

Note that Lemma 1 applies only to error iterates with a bounded Hilbert norm. Our last lemma provides this control for some number of iterations:

**Lemma 3.** *There are constants $(C_1, C_2)$ independent of $n$ such that for any step size $\alpha \in \left(0, \min\{M, \frac{1}{M}\}\right]$, we have*

$$\|\Delta^t\|_{\mathscr{H}} \le C_{\mathscr{H}} \qquad \text{for all iterations } t \le \frac{m}{8M\delta_n^2} \quad (20)$$

*with probability at least $1 - C_1 \exp(-C_2 n\delta_n^2)$, where $C_2 = \max\{\frac{m^2}{\sigma^2}, 1\}$.*

See Section B.3 for the proof of this lemma which also uses Lemma 2.

Taking these lemmas as given, we now complete the proof of the theorem. We first condition on the event $\mathcal{E}$ from Lemma 2, so that we may apply the bound (19). We then fix some iterate $t$ such that $t < \frac{m}{8M\delta_n^2} - 1$, and condition on the event that the bound (20) in Lemma 3 holds, so that we are guaranteed that $\|\Delta^{t+1}\|_{\mathscr{H}} \le C_{\mathscr{H}}$. We then split the analysis into two cases:

**Case 1:**  First, suppose that $\|\Delta^{t+1}\|_n \le \delta_n C_{\mathscr{H}}$. In this case, inequality (12b) holds directly.

**Case 2:** Otherwise, we may assume that $\|\Delta^{t+1}\|_n > \delta_n\|\Delta^{t+1}\|_{\mathscr{H}}$. Applying the bound (19) with the choice $(\widetilde{\Delta}, \Delta) = (\Delta^t, \Delta^{t+1})$ yields

$$\langle \nabla\mathcal{L}(\theta^* + \Delta^t) - \nabla\mathcal{L}_n(\theta^* + \Delta^t), \Delta^{t+1}\rangle \le 4\delta_n\|\Delta^{t+1}\|_n + \frac{m}{c_3}\|\Delta^{t+1}\|_n^2. \tag{21}$$

Substituting inequality (21) back into equation (17) yields

$$\frac{m}{2}\|\Delta^{t+1}\|_n^2 \le \frac{1}{2\alpha}\left\{\|\Delta^t\|_{\mathscr{H}}^2 - \|\Delta^{t+1}\|_{\mathscr{H}}^2\right\} + 4\delta_n\|\Delta^{t+1}\|_n + \frac{m}{c_3}\|\Delta^{t+1}\|_n^2.$$

Re-arranging terms yields the bound

$$\gamma m\|\Delta^{t+1}\|_n^2 \le D^t + 4\delta_n\|\Delta^{t+1}\|_n, \tag{22}$$

where we have introduced the shorthand notation $D^t := \frac{1}{2\alpha}\left\{\|\Delta^t\|_{\mathscr{H}}^2 - \|\Delta^{t+1}\|_{\mathscr{H}}^2\right\}$, as well as $\gamma = \frac{1}{2} - \frac{1}{c_3}$

Equation (22) defines a quadratic inequality with respect to $\|\Delta^{t+1}\|_n$; solving it and making use of the inequality $(a+b)^2 \le 2a^2 + 2b^2$ yields the bound

$$\|\Delta^{t+1}\|_n^2 \le \frac{c\delta_n^2}{\gamma^2 m^2} + \frac{2D^t}{\gamma m}, \tag{23}$$

for some universal constant $c$. By telescoping inequality (23), we find that

$$\frac{1}{T}\sum_{t=1}^{T}\|\Delta^t\|_n^2 \le \frac{c\delta_n^2}{\gamma^2 m^2} + \frac{1}{T}\sum_{t=1}^{T}\frac{2D^t}{\gamma m} \tag{24}$$

$$\le \frac{c\delta_n^2}{\gamma^2 m^2} + \frac{1}{\alpha\gamma m T}[\|\Delta^0\|_{\mathscr{H}}^2 - \|\Delta^T\|_{\mathscr{H}}^2]. \tag{25}$$

By Jensen's inequality, we have

$$\|\bar{f}^T - f^*\|_n^2 = \|\frac{1}{T}\sum_{t=1}^{T}\Delta^t\|_n^2 \le \frac{1}{T}\sum_{t=1}^{T}\|\Delta^t\|_n^2,$$

so that inequality (12b) follows from the bound (24).

On the other hand, by the smoothness assumption, we have

$$\mathcal{L}(\bar{f}^T) - \mathcal{L}(f^*) \le \frac{M}{2}\|\bar{f}^T - f^*\|_n^2,$$

from which inequality (12a) follows.

## A.2  Proof of Corollary 2

The general statement follows directly from Theorem 1. In order to invoke Theorem 1 for the particular cases of LogitBoost and AdaBoost, we need to verify the conditions, i.e. that the $m$-$M$-condition and $\phi'$-boundedness conditions hold for the respective loss function over the ball $\mathbb{B}_{\mathscr{H}}(\theta^*, 2C_{\mathscr{H}})$. The following lemma provides such a guarantee:

**Lemma 4.** With $D := C_{\mathscr{H}} + \|\theta^*\|_{\mathscr{H}}$, the logistic regression cost function satisfies the $m$-$M$-condition with parameters

$$m = \frac{1}{e^{-D} + e^D + 2}, \quad M = \frac{1}{4}, \quad and \quad B = 1.$$

The AdaBoost cost function satisfies the $m$-$M$-condition with parameters

$$m = e^{-D}, \quad M = e^D, \quad and \quad B = e^D.$$

See Section B.4 for the proof of Lemma 4.

$\gamma$-**exponential decay:** If the kernel eigenvalues satisfy a decay condition of the form $\mu_j \le c_1\exp(-c_2 j^\gamma)$, where $c_1, c_2$ are universal constants, the function $\mathcal{R}$ from equation (14) can be upper bounded as

$$\mathcal{R}(\delta) = \sqrt{\frac{2}{n}}\sqrt{\sum_{i=1}^{n}\min\{\delta^2, \mu_j\}} \le \sqrt{\frac{2}{n}}\sqrt{k\delta^2 + \sum_{j=k+1}^{n}c_1 e^{-c_2 j^2}},$$

where $k$ is the smallest integer such that $c_1 \exp(-c_2 k^\gamma) < \delta^2$. Since the localized Gaussian width $\mathcal{G}_n\big(\mathcal{E}_n(\delta, 1)\big)$ can be sandwiched above and below by multiples of $\mathcal{R}(\delta)$, some algebra shows that the critical radius scales as $\delta_n^2 \asymp \frac{n}{\log(n)^{1/\gamma}\sigma^2}$.

Consequently, if we take $T \asymp \frac{\log(n)^{1/\gamma}\sigma^2}{n}$ steps, then Theorem 1 guarantees that the averaged estimator $\bar{\theta}^T$ satisfies the bound

$$\|\bar{\theta}^T - \theta^*\|_n^2 \lesssim \left(\frac{1}{\alpha m} + \frac{1}{m^2}\right) \frac{\log^{1/\gamma} n}{n} \sigma^2,$$

with probability $1 - c_1 \exp(-c_2 m^2 \log^{1/\gamma} n)$.

**$\beta$-polynomial decay:** Now suppose that the kernel eigenvalues satisfy a decay condition of the form $\mu_j \le c_1 j^{-2\beta}$ for some $\beta > 1/2$ and constant $c_1$. In this case, a direct calculation yields the bound

$$\mathcal{R}(\delta) \le \sqrt{\frac{2}{n}}\sqrt{k\delta^2 + c_2 \sum_{j=k+1}^{n} j^{-2}},$$

where $k$ is the smallest integer such that $c_2 k^{-2} < \delta^2$. Combined with upper bound $c_2 \sum_{j=k+1}^{n} j^{-2} \le c_2 \int_{k+1} j^{-2} \le k\delta^2$, we find that the critical radius scales as $\delta_n^2 \asymp n^{-2\beta/(1+2\beta)}$.

Consequently, if we take $T \asymp n^{-2\beta/(1+2\beta)}$ many steps, then Theorem 1 guarantees that the averaged estimator $\bar{\theta}^T$ satisfies the bound

$$\|\bar{\theta}^T - \theta^*\|_n^2 \le \left(\frac{1}{\alpha m} + \frac{1}{m^2}\right)\left(\frac{\sigma^2}{n}\right)^{2\beta/(2\beta+1)},$$

with probability at least $1 - c_1 \exp(-c_2 m^2 (\frac{n}{\sigma^2})^{1/(2\beta+1)})$.

# B    Proof of technical lemmas

## B.1    Proof of Lemma 1

Recalling that $K^\dagger$ denotes the pseudoinverse of $K$, our proof is based on the linear transformation

$$z := n^{-1/2}(K^\dagger)^{1/2}\theta \iff \theta = \sqrt{n}K^{1/2}z.$$

as well as the new function $\mathcal{J}_n(z) := \mathcal{L}_n(\sqrt{n}\sqrt{K}z)$ and its population equivalent $\mathcal{J}(z) := \mathbb{E}\mathcal{J}_n(z)$. Ordinary gradient descent on $\mathcal{J}_n$ with stepsize $\alpha$ takes the form

$$z^{t+1} = z^t - \alpha\nabla\mathcal{J}_n(z^t) = z^t - \alpha\sqrt{n}\sqrt{K}\nabla\mathcal{L}_n(\sqrt{n}\sqrt{K}z^t). \tag{26}$$

If we transform this update on $z$ back to an equivalent one on $\theta$ by multiplying both sides by $\sqrt{n}\sqrt{K}$, we see that ordinary gradient descent on $\mathcal{J}_n$ is equivalent to the kernel boosting update $\theta^{t+1} = \theta^t - \alpha n K\nabla\mathcal{L}_n(\theta^t)$.

Our goal is to analyze the behavior of the update (26) in terms of the population cost $\mathcal{J}(z^t)$. Thus, our problem is one of analyzing a noisy form of gradient descent on the function $\mathcal{J}$, where the noise is induced by the difference between the empirical gradient operator $\nabla\mathcal{J}_n$ and the population gradient operator $\nabla\mathcal{J}$.

Recall that the $\mathcal{L}$ is $M$-smooth by assumption. Since the kernel matrix $K$ has been normalized to have largest eigenvalue at most one, the function $\mathcal{J}$ is also $M$-smooth, whence

$$\mathcal{J}(z^{t+1}) \le \mathcal{J}(z^t) + \langle\nabla\mathcal{J}(z^t), d^t\rangle + \frac{M}{2}\|d^t\|_2^2, \quad \text{where} \quad d^t := z^{t+1} - z^t = -\alpha\nabla\mathcal{J}_n(z^t).$$

Morever, since the function $\mathcal{J}$ is convex, we have $\mathcal{J}(z^*) \ge \mathcal{J}(z^t) + \langle\nabla\mathcal{J}(z^t), z^* - z^t\rangle$, whence

$$\mathcal{J}(z^{t+1}) - \mathcal{J}(z^*) \le \langle\nabla\mathcal{J}(z^t), d^t + z^t - z^*\rangle + \frac{M}{2}\|d^t\|_2^2$$

$$= \langle\nabla\mathcal{J}(z^t), z^{t+1} - z^*\rangle + \frac{M}{2}\|d^t\|_2^2. \tag{27}$$

Now define the difference of the squared errors $V^t := \frac{1}{2}\left\{\|z^t - z^*\|_2^2 - \|z^{t+1} - z^*\|_2^2\right\}$. By some simple algebra, we have

$$
\begin{aligned}
V^t \;=\; \frac{1}{2}\left\{\|z^t - z^*\|_2^2 - \|d^t + z^t - z^*\|_2^2\right\} &= -\langle d^t,\, z^t - z^*\rangle - \frac{1}{2}\|d^t\|_2^2 \\
&= -\langle d^t,\, -d^t + z^{t+1} - z^*\rangle - \frac{1}{2}\|d^t\|_2^2 \\
&= -\langle d^t,\, z^{t+1} - z^*\rangle + \frac{1}{2}\|d^t\|_2^2.
\end{aligned}
$$

Substituting back into equation (27) yields

$$
\mathcal{J}(z^{t+1}) - \mathcal{J}(z^*) \le \frac{1}{\alpha}V^t + \langle \nabla\mathcal{J}(z^t) + \frac{d^t}{\alpha},\, z^{t+1} - z^*\rangle \;=\; \frac{1}{\alpha}V^t + \langle \nabla\mathcal{J}(z^t) - \nabla\mathcal{J}_n(z^t),\, z^{t+1} - z^*\rangle,
$$

where we have used the fact that $\frac{1}{\alpha} \ge M$ by our choice of stepsize $\alpha$.

Finally, we transform back to the original variables $\theta = \sqrt{n}\sqrt{K}z$, using the relation $\nabla\mathcal{J}(z) = \sqrt{n}\sqrt{K}\nabla\mathcal{L}(\theta)$, so as to obtain the bound

$$
\mathcal{L}(\theta^{t+1}) - \mathcal{L}(\theta^*) \le \frac{1}{2\alpha}\left\{\|\Delta^t\|_{\mathscr{H}}^2 - \|\Delta^{t+1}\|_{\mathscr{H}}^2\right\} + \langle \nabla\mathcal{L}(\theta^t) - \nabla\mathcal{L}_n(\theta^t),\, \theta^{t+1} - \theta^*\rangle.
$$

Note that the optimality of $\theta^*$ implies that $\nabla\mathcal{L}(\theta^*) = 0$. Combined with $m$-strong convexity, we are guaranteed that $\frac{m}{2}\|\Delta^{t+1}\|_n^2 \le \mathcal{L}(\theta^{t+1}) - \mathcal{L}(\theta^*)$, and hence

$$
\frac{m}{2}\|\Delta^{t+1}\|_n^2 \le \frac{1}{2\alpha}\left\{\|\Delta^t\|_{\mathscr{H}}^2 - \|\Delta^{t+1}\|_{\mathscr{H}}^2\right\} + \langle \nabla\mathcal{L}(\theta^* + \Delta^t) - \nabla\mathcal{L}_n(\theta^* + \Delta^t),\, \Delta^{t+1}\rangle,
$$

as claimed.

## B.2  Proof of Lemma 2

We split our proof into two cases, depending on whether we are dealing with the least-squares loss $\phi(y,\theta) = \frac{1}{2}(y - \theta)^2$, or a classification loss with uniformly bounded gradient ($\|\phi'\|_\infty \le 1$).

### B.2.1  Least-squares case

The least-squares loss is $m$-strongly convex with $m = M = 1$. Moreover, the difference between the population and empirical gradients can be written as $\nabla\mathcal{L}(\theta^* + \widetilde{\Delta}) - \nabla\mathcal{L}_n(\theta^* + \widetilde{\Delta}) = \frac{\sigma}{n}(w_1, \ldots, w_n)$, where the random variables $\{w_i\}_{i=1}^n$ are i.i.d. and sub-Gaussian with parameter 1. Consequently, we have

$$
|\langle \nabla\mathcal{L}(\theta^* + \widetilde{\Delta}) - \nabla\mathcal{L}_n(\theta^* + \widetilde{\Delta}),\, \Delta\rangle| = \left|\frac{\sigma}{n}\sum_{i=1}^n w_i\Delta(x_i)\right|.
$$

Under these conditions, one can show (see [37] for reference) that

$$
\left|\frac{\sigma}{n}\sum_{i=1}^n w_i\Delta(x_i)\right| \le 2\delta_n\|\Delta\|_n + 2\delta_n^2\|\Delta\|_{\mathscr{H}} + \frac{1}{16}\|\Delta\|_n^2, \tag{28}
$$

which implies that Lemma 2 holds with $c_3 = 16$.

### B.2.2  Gradient-bounded $\phi$-functions

We now turn to the proof of Lemma 2 for gradient bounded $\phi$-functions. First, we claim that it suffices to prove the bound (19) for functions $g \in \partial\mathscr{H}$ and $\|g\|_{\mathscr{H}} = 1$ where $\partial\mathscr{H} := \{f - g \mid f, g \in \mathscr{H}\}$. Indeed, suppose that it holds for all such functions, and that we are given a function $\Delta$ with $\|\Delta\|_{\mathscr{H}} > 1$. By assumption, we can apply the inequality (19) to the new function $g := \Delta/\|\Delta\|_{\mathscr{H}}$, which belongs to $\partial\mathscr{H}$ by nature of the subspace $\mathscr{H} = \overline{\operatorname{span}}\{\mathbb{K}(\cdot, x_i)\}_{i=1}^n$. Applying the bound (19) to $g$ and then multiplying both sides by $\|\Delta\|_{\mathscr{H}}$, we obtain

$$
\begin{aligned}
\langle \nabla\mathcal{L}(\theta^* + \widetilde{\Delta}) - \nabla\mathcal{L}_n(\theta^* + \widetilde{\Delta}),\, \Delta\rangle &\le 2\delta_n\|\Delta\|_n + 2\delta_n^2\|\Delta\|_{\mathscr{H}} + \frac{m}{c_3}\frac{\|\Delta\|_n^2}{\|\Delta\|_{\mathscr{H}}} \\
&\le 2\delta_n\|\Delta\|_n + 2\delta_n^2\|\Delta\|_{\mathscr{H}} + \frac{m}{c_3}\|\Delta\|_n^2,
\end{aligned}
$$

where the second inequality uses the fact that $\|\Delta\|_{\mathscr{H}} > 1$ by assumption.

In order to establish the bound (19) for functions with $\|g\|_{\mathscr{H}} = 1$, we first prove it uniformly over the set $\{g \mid \|g\|_{\mathscr{H}} = 1, \quad \|g\|_n \le t\}$, where $t > 1$ is a fixed radius (of course, we restrict our attention to those radii $t$

for which this set is non-empty.) We then extend the argument to one that is also uniform over the choice of $t$ by a "peeling" argument.

Define the random variable

$$\mathcal{Z}_n(t) := \sup_{\Delta, \widetilde{\Delta} \in \mathcal{E}(t,1)} \langle \nabla \mathcal{L}(\theta^* + \widetilde{\Delta}) - \nabla \mathcal{L}_n(\theta^* + \widetilde{\Delta}), \Delta \rangle. \tag{29}$$

The following two lemmas, respectively, bound the mean of this random variable, and its deviations above the mean:

**Lemma 5.** *For any $t > 0$, the mean is upper bounded as*

$$\mathbb{E}\mathcal{Z}_n(t) \leq \sigma \mathcal{G}_n(\mathcal{E}(t,1)), \tag{30}$$

*where $\sigma := 2M + 4C_{\mathscr{H}}$.*

**Lemma 6.** *There are universal constants $(c_1, c_2)$ such that*

$$\mathbb{P}\Big[\mathcal{Z}_n(t) \geq \mathbb{E}\mathcal{Z}_n(t) + \alpha\Big] \leq c_1 \exp\Big(-\frac{c_2 n \alpha^2}{t^2}\Big). \tag{31}$$

See Appendices B.2.3 and B.2.4 for the proofs of these two claims.

Equipped with Lemmas 5 and 6, we now prove inequality (19). We divide our argument into two cases:

**Case $t = \delta_n$:** We first prove inequality (19) for $t = \delta_n$. From Lemma 5, we have

$$\mathbb{E}\mathcal{Z}_n(\delta_n) \leq \sigma \mathcal{G}_n(\mathcal{E}(\delta_n, 1)) \overset{(i)}{\leq} \delta_n^2, \tag{32}$$

where inequality (i) follows from the definition of $\delta_n$ in inequality (11). Setting $\alpha = \delta_n^2$ in expression (31) yields

$$\mathbb{P}\Big[\mathcal{Z}_n(\delta_n) \geq 2\delta_n^2\Big] \leq c_1 \exp\left(-c_2 n \delta_n^2\right), \tag{33}$$

which establishes the claim for $t = \delta_n$.

**Case $t > \delta_n$:** On the other hand, for any $t > \delta_n$, we have

$$\mathbb{E}\mathcal{Z}_n(t) \overset{(i)}{\leq} \sigma \mathcal{G}_n(\mathcal{E}(t,1)) \overset{(ii)}{\leq} t\sigma \frac{\mathcal{G}_n(\mathcal{E}(t,1))}{t} \leq t\delta_n,$$

where step (i) follows from Lemma 5, and step (ii) follows because the function $u \mapsto \frac{\mathcal{G}_n(\mathcal{E}(u,1))}{u}$ is non-increasing on the positive real line. (This non-increasing property is a direct consequence of the star-shaped nature of $\partial\mathscr{H}$.) Finally, using this upper bound on expression $\mathbb{E}\mathcal{Z}_n(\delta_n)$ and setting $\alpha = t^2 m / (4c_3)$ in the tail bound (31) yields

$$\mathbb{P}\Big[\mathcal{Z}_n(t) \geq t\delta_n + \frac{t^2 m}{4c_3}\Big] \leq c_1 \exp\left(-c_2 n m^2 t^2\right). \tag{34}$$

Note that the precise values of the universal constants $c_2$ may change from line to line throughout this section.

**Peeling argument** Equipped with the tail bounds (33) and (34), we are now ready to complete the peeling argument. Let $\mathcal{A}$ denote the event that the bound (19) is violated for some function $g \in \partial\mathscr{H}$ with $\|g\|_{\mathscr{H}} = 1$. For real numbers $0 \leq a < b$, let $\mathcal{A}(a,b)$ denote the event that it is violated for some function such that $\|g\|_n \in [a,b]$, and $\|g\|_{\mathscr{H}} = 1$. For $k = 0, 1, 2, \ldots$, define $t_k = 2^k \delta_n$. We then have the decomposition $\mathcal{E} = (0, t_0) \cup (\bigcup_{k=0}^{\infty} \mathcal{A}(t_k, t_{k+1}))$ and hence by union bound,

$$\mathbb{P}[\mathcal{E}] \leq \mathbb{P}[\mathcal{A}(0, \delta_n)] + \sum_{k=1}^{\infty} \mathbb{P}[\mathcal{A}(t_k, t_{k+1})]. \tag{35}$$

From the bound (33), we have $\mathbb{P}[\mathcal{A}(0, \delta_n)] \leq c_1 \exp\left(-c_2 n \delta_n^2\right)$. On the other hand, suppose that $\mathcal{A}(t_k, t_{k+1})$ holds, meaning that there exists some function $g$ with $\|g\|_{\mathscr{H}} = 1$ and $\|g\|_n \in [t_k, t_{k+1}]$ such that

$$\langle \nabla \mathcal{L}(\theta^* + \widetilde{\Delta}) - \nabla \mathcal{L}_n(\theta^* + \widetilde{\Delta}), g \rangle > 2\delta_n \|g\|_n + 2\delta_n^2 + \frac{m}{c_3}\|g\|_n^2$$

$$\overset{(i)}{\geq} 2\delta_n t_k + 2\delta_n^2 + \frac{m}{c_3}t_k^2$$

$$\overset{(ii)}{\geq} \delta_n t_{k+1} + 2\delta_n^2 + \frac{m}{4c_3}t_{k+1}^2,$$

where step (i) uses the $\|g\|_n \geq t_k$ and step (ii) uses the fact that $t_{k+1} = 2t_k$. This lower bound implies that $\mathcal{Z}_n(t_{k+1}) > t_{k+1}\delta_n + \frac{t_{k+1}^2 m}{4c_3}$ and applying the tail bound (34) yields

$$\mathbb{P}(\mathcal{A}(t_k, t_{k+1})) \leq \mathbb{P}(\mathcal{Z}_n(t_{k+1}) > t_{k+1}\delta_n + \frac{t_{k+1}^2 m}{4c_3}) \leq \exp\left(-c_2 nm^2 2^{2k+2}\delta_n^2\right).$$

Substituting this inequality and our earlier bound (33) into equation (35) yields

$$\mathbb{P}(\mathcal{E}) \leq c_1 \exp(-c_2 nm^2\delta_n^2),$$

where the reader should recall that the precise values of universal constants may change from line-to-line. Since $\sigma^2 > 1$ by definition, this concludes the proof of Lemma 2.

### B.2.3   Proof of Lemma 5

Recalling the definitions (1) and (2) of $\mathcal{L}$ and $\mathcal{L}_n$, we can write

$$\mathcal{Z}_n(t) = \sup_{\Delta, \widetilde{\Delta} \in \mathcal{E}(t,1)} \frac{1}{n} \sum_{i=1}^{n} (\phi'(y_i, \theta_i^* + \widetilde{\Delta}_i) - \mathbb{E}\phi'(y_i, \theta_i^* + \widetilde{\Delta}_i))\Delta_i$$

Note that the vectors $\Delta$ and $\widetilde{\Delta}$ contain function values of the form $f(x_i) - f^*(x_i)$ for functions $f \in \mathbb{B}_{\mathscr{H}}(f^*, 2C_{\mathscr{H}})$. Recall that the kernel function is bounded uniformly by one. Consequently, for any function $f \in \mathbb{B}_{\mathscr{H}}(f^*, 2C_{\mathscr{H}})$, we have

$$|f(x) - f^*(x)| = |\langle f - f^*, \mathbb{K}(\cdot, x)\rangle_{\mathscr{H}}| \leq \|f - f^*\|_{\mathscr{H}}\|\mathbb{K}(\cdot, x)\|_{\mathscr{H}} \leq 2C_{\mathscr{H}}.$$

Thus, we can restrict our attention to vectors $\Delta, \widetilde{\Delta}$ with $\|\Delta\|_\infty, \|\widetilde{\Delta}\|_\infty \leq 2C_{\mathscr{H}}$ from hereonwards.

Letting $\{\varepsilon_i\}_{i=1}^{n}$ denote an i.i.d. sequence of Rademacher variables, define the symmetrized variable

$$\tilde{\mathcal{Z}}_n(t) := \sup_{\Delta, \widetilde{\Delta} \in \mathcal{E}(t,1)} \frac{1}{n} \sum_{i=1}^{n} \varepsilon_i \phi'(y_i, \theta_i^* + \widetilde{\Delta}_i)\, \Delta_i. \tag{36}$$

By a standard symmetrization argument [33], we have $\mathbb{E}_y[\mathcal{Z}_n(t)] \leq 2\mathbb{E}_{y,\epsilon}[\tilde{\mathcal{Z}}_n(t)]$. Moreover, since

$$\phi'(y_i, \theta_i^* + \widetilde{\Delta}_i)\, \Delta_i \leq \frac{1}{2}\left(\phi'(y_i, \theta_i^* + \widetilde{\Delta}_i)\right)^2 + \frac{1}{2}\Delta_i^2$$

we have

$$\mathbb{E}\mathcal{Z}_n(t) \leq \mathbb{E} \sup_{\widetilde{\Delta} \in \mathcal{E}(t,1)} \frac{1}{n} \sum_{i=1}^{n} \varepsilon_i \left(\phi'(y_i, \theta_i^* + \widetilde{\Delta}_i)\right)^2 \; + \; \mathbb{E} \sup_{\Delta \in \mathcal{E}(t,1)} \frac{1}{n} \sum_{i=1}^{n} \varepsilon_i \Delta_i^2$$

$$\leq \underbrace{2\,\mathbb{E} \sup_{\widetilde{\Delta} \in \mathcal{E}(t,1)} \frac{1}{n} \sum_{i=1}^{n} \varepsilon_i \phi'(y_i, \theta_i^* + \widetilde{\Delta}_i)}_{T_1} \; + \; 4C_{\mathscr{H}}\, \underbrace{\mathbb{E} \sup_{\Delta \in \mathcal{E}(t,1)} \frac{1}{n} \sum_{i=1}^{n} \varepsilon_i \Delta_i}_{T_2},$$

where the second inequality follows by applying the Rademacher contraction inequality [22], using the fact that $\|\phi'\|_\infty \leq 1$ for the first term, and $\|\Delta\|_\infty \leq 2C_{\mathscr{H}}$ for the second term.

Focusing first on the term $T_1$, since $\mathbb{E}[\varepsilon_i \phi'(y_i, \theta_i^*)] = 0$, we have

$$T_1 = \mathbb{E} \sup_{\widetilde{\Delta} \in \mathcal{E}(t,1)} \frac{1}{n} \sum_{i=1}^{n} \varepsilon_i \underbrace{\left(\phi'(y_i, \theta_i^* + \widetilde{\Delta}_i) - \phi'(y_i; \theta_i^*)\right)}_{\varphi_i(\widetilde{\Delta}_i)}$$

$$\overset{(i)}{\leq} M\mathbb{E} \sup_{\widetilde{\Delta} \in \mathcal{E}(t,1)} \frac{1}{n} \sum_{i=1}^{n} \varepsilon_i \widetilde{\Delta}_i$$

$$\overset{(ii)}{\leq} M\mathcal{G}_n(\mathcal{E}(t,1)),$$

where step (i) follows since each function $\varphi_i$ is $M$-Lipschitz by assumption; and step (ii) follows since the Gaussian complexity upper bounds the Rademacher complexity. Similarly, we have

$$T_2 \leq \mathcal{G}_n(\mathcal{E}(t,1)),$$

and putting together the pieces yields the claim.

### B.2.4 Proof of Lemma 6

Recall the definition (36) of the symmetrized variable $\tilde{\mathcal{Z}}_n$. By a standard symmetrization argument [33], there are universal constants $c_1, c_2$ such that

$$\mathbb{P}\Big[\mathcal{Z}_n(t) \geq \mathbb{E}\mathcal{Z}_n[t] + c_1\alpha\Big] \leq c_2\mathbb{P}\Big[\tilde{\mathcal{Z}}_n(t) \geq \mathbb{E}\tilde{\mathcal{Z}}_n[t] + \alpha\Big].$$

Since $\{\varepsilon_i\}_{i=1}^n$ are $\{y_i\}_{i=1}^n$ are independent, we can study $\tilde{\mathcal{Z}}_n(t)$ conditionally on $\{y_i\}_{i=1}^n$. Viewed as a function of $\{\varepsilon_i\}_{i=1}^n$, the function $\tilde{\mathcal{Z}}_n(t)$ is convex and Lipschitz with respect to the Euclidean norm with parameter

$$L^2 := \sup_{\Delta, \tilde{\Delta} \in \mathcal{E}(t,1)} \frac{1}{n^2} \sum_{i=1}^n \Big(\phi'(y_i, \theta_i^* + \tilde{\Delta}_i) \, \Delta_i\Big)^2 \leq \frac{t^2}{n},$$

where we have used the facts that $\|\phi'\|_\infty \leq 1$ and $\|\Delta\|_n \leq t$. By Ledoux's concentration for convex and Lipschitz functions [21], we have

$$\mathbb{P}\Big[\tilde{\mathcal{Z}}_n(t) \geq \mathbb{E}\tilde{\mathcal{Z}}_n[t] + \alpha \mid \{y_i\}_{i=1}^n\Big] \leq c_3 \exp\Big(-c_4 \frac{n\alpha^2}{t^2}\Big).$$

Since the right-hand side does not involve $\{y_i\}_{i=1}^n$, the same bound holds unconditionally over the randomness in both the Rademacher variables and the sequence $\{y_i\}_{i=1}^n$. Consequently, the claimed bound (31) follows, with suitable redefinitions of the universal constants.

## B.3 Proof of Lemma 3

We first require an auxiliary lemma, which we state and prove in the following section. We then prove Lemma 3 in Section B.3.2.

### B.3.1 An auxiliary lemma

The following result relates the Hilbert norm of the error to the difference between the empirical and population gradients:

**Lemma 7.** *For any convex and differentiable loss function $\mathcal{L}$, the kernel boosting error $\Delta^{t+1} := \theta^{t+1} - \theta^*$ satisfies the bound*

$$\|\Delta^{t+1}\|_{\mathscr{H}}^2 \leq \|\Delta^t\|_{\mathscr{H}}\|\Delta^{t+1}\|_{\mathscr{H}} + \alpha\langle \nabla\mathcal{L}(\theta^* + \Delta^t) - \nabla\mathcal{L}_n(\theta^* + \Delta^t), \, \Delta^{t+1}\rangle. \tag{37}$$

*Proof.* Recall that $\|\Delta^t\|_{\mathscr{H}}^2 = \|\theta^t - \theta^*\|_{\mathscr{H}}^2 = \|z^t - z^*\|_2^2$ by definition of the Hilbert norm. Let us define the population update operator $G$ on the population function $\mathcal{J}$ and the empirical update operator $G_n$ on $\mathcal{J}_n$ as

$$G(z^t) := z^t - \alpha\nabla\mathcal{J}(\sqrt{n}\sqrt{K}z^t), \quad \text{and} \quad z^{t+1} := G_n(z^t) = z^t - \alpha\nabla\mathcal{J}_n(\sqrt{n}\sqrt{K}z^t). \tag{38}$$

Since $\mathcal{J}$ is convex and smooth, it follows from standard arguments in convex optimization that $G$ is a nonexpansive operator—viz.

$$\|G(x) - G(y)\|_2 \leq \|x - y\|_2 \qquad \text{for all } x, y \in \mathcal{C}. \tag{39}$$

In addition, we note that the vector $z^*$ is a fixed point of $G$—that is, $G(z^*) = z^*$. From these ingredients, we have

$$\|\Delta^{t+1}\|_{\mathscr{H}}^2 = \langle z^{t+1} - z^*, \, G_n(z^t) - G(z^t) + G(z^t) - z^*\rangle$$

$$\overset{(i)}{\leq} \|z^{t+1} - z^*\|_2\|G(z^t) - G(z^*)\|_2 + \alpha\langle\sqrt{n}\sqrt{K}[\nabla\mathcal{L}(\theta^* + \Delta^t) - \nabla\mathcal{L}_n(\theta^* + \Delta^t)], \, z^{t+1} - z^*\rangle$$

$$\overset{(ii)}{\leq} \|\Delta^{t+1}\|_{\mathscr{H}}\|\Delta^t\|_{\mathscr{H}} + \alpha\langle\nabla\mathcal{L}(\theta^* + \Delta^t) - \nabla\mathcal{L}_n(\theta^* + \Delta^t), \, \Delta^{t+1}\rangle$$

where step (i) follows by applying the Cauchy-Schwarz to control the inner product, and step (ii) follows since $\Delta^{t+1} = \sqrt{n}\sqrt{K}(z^{t+1} - z^*)$, and the square root kernel matrix $\sqrt{K}$ is symmetric. $\qquad\square$

### B.3.2 Proof of Lemma 3

We now prove Lemma 3, in which we make use of Lemma 1 and Lemma 2 combined with Lemma 7.

In order to prove inequality (20), we follow an inductive argument. Instead of proving (20) directly, we prove a slightly stronger relation which implies it, i.e.

$$\max\{1, \|\Delta^t\|_{\mathscr{H}}^2\} \leq \max\{1, \|\Delta^0\|_{\mathscr{H}}^2\} + t\delta_n^2 \frac{4M}{\widetilde{\gamma}m} \tag{40}$$

for constants $\widetilde{\gamma}, c_3$ such that

$$\widetilde{\gamma} := \frac{1}{32} - \frac{1}{4c_3} = 1/C_{\mathcal{H}}^2. \tag{41}$$

We claim that it suffices to prove that the error iterates $\Delta^{t+1}$ satisfy the inequality (40). Indeed, if we take inequality (40) as given, then we have

$$\|\Delta^t\|_{\mathcal{H}}^2 \leq \max\{1, \|\Delta^0\|_{\mathcal{H}}^2\} + \frac{1}{2\widetilde{\gamma}} \leq C_{\mathcal{H}}^2,$$

where we used the definition $C_{\mathcal{H}}^2 = 2\max\{\|\theta^*\|_{\mathcal{H}}^2, 32\}$. Thus, it suffices to focus our attention on proving inequality (40).

For $t = 0$, it is trivially true. Now let us assume inequality (40) holds for some $t \leq \frac{m}{8M\delta_n^2}$, and then prove that it also holds for step $t + 1$.

If $\|\Delta^{t+1}\|_{\mathcal{H}} < 1$, then inequality (40) follows directly. Therefore, we can assume in the following without loss of generality that $\|\Delta^{t+1}\|_{\mathcal{H}} \geq 1$.

We break down the proof of this induction into two steps:

- first showing that $\|\Delta^{t+1}\|_{\mathcal{H}} \leq 2C_{\mathcal{H}}$ so that Lemma 2 is applicable.
- second, showing that the bound (40) holds and thus in fact $\|\Delta^{t+1}\|_{\mathcal{H}} \leq C_{\mathcal{H}}$.

Throughout the proof, we condition on the event $\mathcal{E}$ and $\mathcal{E}_0 := \{\frac{1}{\sqrt{n}}\|y - \mathbb{E}[y \mid x]\|_2 \leq \sqrt{2}\sigma\}$. Lemma 2 guarantees that $\mathbb{P}(\mathcal{E}^c) \leq c_1 \exp(-c_2\frac{m^2 n \delta_n^2}{\sigma^2})$ whereas $\mathbb{P}(\mathcal{E}_0) \geq 1 - \mathrm{e}^{-n}$ follows from the fact that $Y^2$ is sub-exponential with parameter $\sigma^2 n$ and applying Hoeffding's inequality. Putting things together we obtain the upper bound of the complement event

$$\mathbb{P}(\mathcal{E}^c \cup \mathcal{E}_0^c) \leq 2c_1 \exp(-C_2 n \delta_n^2)$$

with $C_2 = \max\{\frac{m^2}{\sigma^2}, 1\}$.

**Showing that $\|\Delta^{t+1}\|_{\mathcal{H}} \leq 2C_{\mathcal{H}}$**   In this step, we assume that inequality (40) holds at step $t$, and show that $\|\Delta^{t+1}\|_{\mathcal{H}} \leq 2C_{\mathcal{H}}$. Recalling that $z := \frac{(K^\dagger)^{1/2}}{\sqrt{n}}\theta$, our update can be written as

$$z^{t+1} - z^* = z^t - \alpha\sqrt{n}\sqrt{K}\nabla\mathcal{L}(\theta^t) - z^* + \alpha\sqrt{n}\sqrt{K}(\nabla\mathcal{L}_n(\theta^t) - \nabla\mathcal{L}(\theta^t)).$$

Applying the triangle inequality yields the bound

$$\|z^{t+1} - z^*\|_2 \leq \|\underbrace{z^t - \alpha\sqrt{n}\sqrt{K}\nabla\mathcal{L}(\theta^t)}_{G(z^t)} - z^*\|_2 + \|\alpha\sqrt{n}\sqrt{K}(\nabla\mathcal{L}_n(\theta^t) - \nabla\mathcal{L}(\theta^t))\|_2$$

where the population update operator $G$ was previously defined (38), and observed to be non-expansive (39). From this non-expansiveness, we find that

$$\|z^{t+1} - z^*\|_2 \leq \|z^t - z^*\|_2 + \|\alpha\sqrt{n}\sqrt{K}(\nabla\mathcal{L}_n(\theta^t) - \nabla\mathcal{L}(\theta^t))\|_2,$$

Note that the $\ell_2$ norm of $z$ corresponds to the Hilbert norm of $\theta$. This implies

$$\|\Delta^{t+1}\|_{\mathcal{H}} \leq \|\Delta^t\|_{\mathcal{H}} + \underbrace{\|\alpha\sqrt{n}\sqrt{K}(\nabla\mathcal{L}_n(\theta^t) - \nabla\mathcal{L}(\theta^t))\|_2}_{:=T}$$

Observe that because of uniform boundedness of the kernel by one, the quantity $T$ can be bounded as

$$T \leq \alpha\sqrt{n}\|\nabla\mathcal{L}_n(\theta^t) - \nabla\mathcal{L}(\theta^t))\|_2 = \alpha\sqrt{n}\frac{1}{n}\|v - \mathbb{E}v\|_2,$$

where we have define the vector $v \in \mathbb{R}^n$ with coordinates $v_i := \phi'(y_i, \theta_i^t)$. For functions $\phi$ satisfying the gradient boundedness and $m - M$ condition, since $\theta^t \in \mathbb{B}_{\mathcal{H}}(\theta^*, C_{\mathcal{H}})$, each coordinate of the vectors $v$ and $\mathbb{E}v$ is bounded by 1 in absolute value. We consequently have

$$T \leq \alpha \leq C_{\mathcal{H}},$$

where we have used the fact that $\alpha \leq m/M < 1 \leq \frac{C_{\mathcal{H}}}{2}$. For least-squares $\phi$ we instead have

$$T \leq \alpha\frac{\sqrt{n}}{n}\|y - \mathbb{E}[y \mid x]\|_2 =: \frac{\alpha}{\sqrt{n}}Y \leq \sqrt{2}\sigma \leq C_{\mathcal{H}}$$

conditioned on the event $\mathcal{E}_0 := \{\frac{1}{\sqrt{n}}\|y - \mathbb{E}[y \mid x]\|_2 \leq \sqrt{2}\sigma\}$. Since $Y^2$ is sub-exponential with parameter $\sigma^2 n$ it follows by Hoeffding's inequality that $\mathbb{P}(\mathcal{E}_0) \geq 1 - \mathrm{e}^{-n}$.

Putting together the pieces yields that $\|\Delta^{t+1}\|_{\mathcal{H}} \leq 2C_{\mathcal{H}}$, as claimed.

**Completing the induction step** We are now ready to complete the induction step for proving inequality (40) using Lemma 1 and Lemma 2 since $\|\Delta^{t+1}\|_{\mathscr{H}} \geq 1$. We split the argument into two cases separately depending on whether or not $\|\Delta^{t+1}\|_{\mathscr{H}}\delta_n \geq \|\Delta^{t+1}\|_n$. In general we can assume that $\|\Delta^{t+1}\|_{\mathscr{H}} > \|\Delta^t\|_{\mathscr{H}}$, otherwise the induction inequality (40) satisfies trivially.

**Case 1:** When $\|\Delta^{t+1}\|_{\mathscr{H}}\delta_n \geq \|\Delta^{t+1}\|_n$, inequality (19) implies that

$$\langle \nabla \mathcal{L}(\theta^* + \widetilde{\Delta}) - \nabla \mathcal{L}_n(\theta^* + \widetilde{\Delta}), \Delta^{t+1} \rangle \leq 4\delta_n^2 \|\Delta^{t+1}\|_{\mathscr{H}} + \frac{m}{c_3}\|\Delta^{t+1}\|_n^2, \tag{42}$$

Combining Lemma 7 and inequality (42), we obtain

$$\|\Delta^{t+1}\|_{\mathscr{H}}^2 \leq \|\Delta^t\|_{\mathscr{H}}\|\Delta^{t+1}\|_{\mathscr{H}} + 4\alpha\delta_n^2\|\Delta^{t+1}\|_{\mathscr{H}} + \alpha\frac{m}{c_3}\|\Delta^{t+1}\|_n^2$$

$$\implies \|\Delta^{t+1}\|_{\mathscr{H}} \leq \frac{1}{1 - \alpha\delta_n^2\frac{m}{c_3}}\left[\|\Delta^t\|_{\mathscr{H}} + 4\alpha\delta_n^2\right], \tag{43}$$

where the last inequality uses the fact that $\|\Delta^{t+1}\|_n \leq \delta_n\|\Delta^{t+1}\|_{\mathscr{H}}$.

**Case 2:** When $\|\Delta^{t+1}\|_{\mathscr{H}}\delta_n < \|\Delta^{t+1}\|_n$, we use our assumption $\|\Delta^{t+1}\|_{\mathscr{H}} \geq \|\Delta^t\|_{\mathscr{H}}$ together with Lemma 7 and inequality (19) which guarantee that

$$\|\Delta^{t+1}\|_{\mathscr{H}}^2 \leq \|\Delta^t\|_{\mathscr{H}}^2 + 2\alpha\langle \nabla\mathcal{L}(\theta^* + \Delta^t) - \nabla\mathcal{L}_n(\theta^* + \Delta^t), \Delta^{t+1}\rangle$$

$$\leq \|\Delta^t\|_{\mathscr{H}}^2 + 8\alpha\delta_n\|\Delta^{t+1}\|_n + 2\alpha\frac{m}{c_3}\|\Delta^{t+1}\|_n^2.$$

Using the elementary inequality $2ab \leq a^2 + b^2$, we find that

$$\|\Delta^{t+1}\|_{\mathscr{H}}^2 \leq \|\Delta^t\|_{\mathscr{H}}^2 + 8\alpha\left[m\widetilde{\gamma}\|\Delta^{t+1}\|_n^2 + \frac{1}{4\widetilde{\gamma}m}\delta_n^2\right] + 2\alpha\frac{m}{c_3}\|\Delta^{t+1}\|_n^2$$

$$\leq \|\Delta^t\|_{\mathscr{H}}^2 + \alpha\frac{m}{4}\|\Delta^{t+1}\|_n^2 + \frac{2\alpha\delta_n^2}{\widetilde{\gamma}m}, \tag{44}$$

where in the final step, we plug in the constants $\widetilde{\gamma}, c_3$ which satisfy equation (41).

Now Lemma 1 implies that

$$\frac{m}{2}\|\Delta^{t+1}\|_n^2 \leq D^t + 4\|\Delta^{t+1}\|_n\delta_n + \frac{m}{c_3}\|\Delta^{t+1}\|_n^2$$

$$\overset{(i)}{\leq} D^t + 4\left[\widetilde{\gamma}m\|\Delta^{t+1}\|_n^2 + \frac{1}{4\widetilde{\gamma}m}\delta_n^2\right] + \frac{m}{c_3}\|\Delta^{t+1}\|_n^2,$$

where step (i) again uses $2ab \leq a^2 + b^2$. Thus, we have $\frac{m}{4}\|\Delta^{t+1}\|_n^2 \leq D^t + \frac{1}{\widetilde{\gamma}m}\delta_n^2$. Together with expression (44), we find that

$$\|\Delta^{t+1}\|_{\mathscr{H}}^2 \leq \|\Delta^t\|_{\mathscr{H}}^2 + \frac{1}{2}(\|\Delta^t\|_{\mathscr{H}}^2 - \|\Delta^{t+1}\|_{\mathscr{H}}^2) + \frac{4\alpha}{\widetilde{\gamma}m}\delta_n^2$$

$$\implies \|\Delta^{t+1}\|_{\mathscr{H}}^2 \leq \|\Delta^t\|_{\mathscr{H}}^2 + \frac{4\alpha}{\widetilde{\gamma}m}\delta_n^2. \tag{45}$$

**Combining the pieces:** By combining the two previous cases, we arrive at the bound

$$\max\left\{1, \|\Delta^{t+1}\|_{\mathscr{H}}^2\right\} \leq \max\left\{1, \kappa^2(\|\Delta^t\|_{\mathscr{H}} + 4\alpha\delta_n^2)^2, \|\Delta^t\|_{\mathscr{H}}^2 + \frac{4M}{\widetilde{\gamma}m}\delta_n^2\right\}, \tag{46}$$

where $\kappa := \frac{1}{(1 - \alpha\delta_n^2\frac{m}{c_3})}$ and we used that $\alpha \leq \min\{\frac{1}{M}, M\}$.

Now it is only left for us to show that with the constant $c_3$ chosen such that $\widetilde{\gamma} = \frac{1}{32} - \frac{1}{4c_3} = 1/C_{\mathscr{H}}^2$, we have

$$\kappa^2(\|\Delta^t\|_{\mathscr{H}} + 4\alpha\delta_n^2)^2 \leq \|\Delta^t\|_{\mathscr{H}}^2 + \frac{4M}{\widetilde{\gamma}m}\delta_n^2.$$

Define the function $f : (0, C_{\mathscr{H}}] \to \mathbb{R}$ via $f(\xi) := \kappa^2(\xi + 4\alpha\delta_n^2)^2 - \xi^2 - \frac{4M}{\widetilde{\gamma}m}\delta_n^2$. Since $\kappa \geq 1$, in order to conclude that $f(\xi) < 0$ for all $\xi \in (0, C_{\mathscr{H}}]$, it suffices to show that $\mathrm{argmin}_{x\in\mathbb{R}} f(x) < 0$ and $f(C_{\mathscr{H}}) < 0$. The former is obtained by basic algebra and follows directly from $\kappa \geq 1$. For the latter, since $\widetilde{\gamma} = \frac{1}{32} - \frac{1}{4c_3} = 1/C_{\mathscr{H}}^2, \alpha < \frac{1}{M}$ and $\delta_n^2 \leq \frac{M^2}{m^2}$ it thus suffices to show

$$\frac{1}{(1 - \frac{M}{8m})^2} \leq \frac{4M}{m} + 1$$

Since $(4x + 1)(1 - \frac{x}{8})^2 \geq 1$ for all $x \leq 1$ and $\frac{m}{M} \leq 1$, we conclude that $f(C_{\mathscr{H}}) < 0$.

Now that we have established $\max\{1, \|\Delta^{t+1}\|_{\mathscr{H}}^2\} \leq \max\{1, \|\Delta^t\|_{\mathscr{H}}^2\} + \frac{4M}{\gamma m}\delta_n^2$, the induction step (40) follows. which completes the proof of Lemma 3.

## B.4 Proof of Lemma 4

Recall that the LogitBoost algorithm is based on logistic loss $\phi(y, \theta) = \ln(1 + e^{-y\theta})$, whereas the AdaBoost algorithm is based on the exponential loss $\phi(y, \theta) = \exp(-y\theta)$. We now verify the $m$-$M$-condition for these two losses with the corresponding parameters specified in Lemma 4.

### B.4.1 $m$-$M$-condition for logistic loss

The first and second derivatives are given by

$$\frac{\partial \phi(y, \theta)}{\partial \theta} = \frac{-ye^{-y\theta}}{1 + e^{-y\theta}}, \qquad \text{and} \qquad \frac{\partial^2 \phi(y, \theta)}{(\partial \theta)^2} = \frac{y^2}{(e^{-y\theta/2} + e^{y\theta/2})^2}.$$

It is easy to check that $|\frac{\partial \phi(y,\theta)}{\partial \theta}|$ is uniformly bounded by $B = 1$.

Turning to the second derivative, recalling that $y \in \{-1, +1\}$, it is straightforward to show that

$$\max_{y \in \{-1, +1\}} \sup_{\theta} \frac{y^2}{(e^{-y\theta/2} + e^{y\theta/2})^2} \leq \frac{1}{4},$$

which implies that $\frac{\partial \phi(y,\theta)}{\partial \theta}$ is a $1/4$-Lipschitz function of $\theta$, i.e. with $M = 1/4$.

Our final step is to compute a value for $m$ by deriving a uniform lower bound on the Hessian. For this step, we need to exploit the fact that $\theta = f(x)$ must arise from a function $f$ such that $\|f\|_{\mathscr{H}} \leq D := C_{\mathscr{H}} + \|\theta^*\|_{\mathscr{H}}$. Since $\sup_x \mathbb{K}(x, x) \leq 1$ by assumption, the reproducing relation for RKHS then implies that $|f(x)| \leq D$. Combining this inequality with the fact that $y \in \{-1, 1\}$, it suffices to lower the bound the quantity

$$\min_{y \in \{-1, +1\}} \min_{|\theta| \leq D} \left| \frac{\partial^2 \phi(y, \theta)}{(\partial \theta)^2} \right| = \min_{|y| \leq 1} \min_{|\theta| \leq D} \frac{y^2}{(e^{-y\theta/2} + e^{y\theta/2})^2} \geq \underbrace{\frac{1}{e^{-D} + e^D + 2}}_{m},$$

which completes the proof for the logistic loss.

### B.4.2 $m$-$M$-condition for AdaBoost

The AdaBoost algorithm is based on the cost function $\phi(y, \theta) = e^{-y\theta}$, which has first and second derivatives (with respect to its second argument) given by

$$\frac{\partial \phi(y, \theta)}{\partial \theta} = -ye^{-y\theta}, \qquad \text{and} \qquad \frac{\partial^2 \phi(y, \theta)}{(\partial \theta)^2} = e^{-y\theta}.$$

As in the preceding argument for logistic loss, we have the bound $|y| \leq 1$ and $|\theta| \leq D$. By inspection, the absolute value of the first derivative is uniformly bounded $B := e^D$, whereas the second derivative always lies in the interval $[m, M]$ with $M := e^D$ and $m := e^{-D}$, as claimed.