[Reviews · NeurIPS 2017]

Reviewer 1



The authors proved an upper bound of early stopping strategy for kernel boosting algorithm and determined the number of iteration steps to achieve minimax optimal rate. The upper bounds are stated in terms of localized Gaussian complexity and lead to fast rates for kernels with fast decaying eigenvalues. These results are novel and promising. A small concern is on the the authors' claim that it is straightforward to extend the results for empirical L2 norm to population L2 norm. I have a little bit reservation on this before seeing the detailed proof.

Reviewer 2



The paper establishes minimax optimal rates of estimation for early stopping of kernel boosting, where the stopping index is expressed in terms of localized Rademacher/Gaussian complexities. Overall, this paper is well written and closes a gap in the literature: while lower bounds in terms of localized complexity measures are known, a similar result for the choice of the nb. of iterations (and therefore upper bounds) was unknown. Some comments: ***Remark after Th. 1 about random design setting should be more explained (at least for the not initiated reader); applies (11b) to squared population L^2-norm with f^T or \bar f^T ? ***l. 188: I think Yang et al. [30] state a lower bound involving the critical radius delta_n defined as the smallest positive solution such that R(delta)<=delta^2/sigma , with R(delta) given in (6) (rather than the worst-case constant sup delta^2_n; or are these two numbers equivalent ?). ***l.223/224: could you provide a reference? ***a possible weak spot: numerical experiments are provided only for the error involving f^T in distinction to Th. 1. The authors remark in l.282/283 that a rigorous proof is missing (on the other hand, based on this simulations one can indeed strongly suspect that the same result holds for f^T as well) A few very minor remarks about typesetting: *** l. 155 ff. --> the function class {\mathcal E} and also delta depend on n, which could be indicated for better understanding; same for the effective noise level sigma for least squares *** l.158: "Gaussian complexity ... at scale sigma"-->should be "delta"